# Patterns of sedentary behaviour in adults with acute insomnia derived from actigraphy data

**Sunita Rani**[1], **Sergiy Shelyag**[1,2], **Maia Angelova**[1]*

**1** School of IT, Deakin University, Melbourne, Victoria, Australia, **2** College of Science and Engineering, Flinders University, Adelaide, South Australia, Australia

* maia.a@deakin.edu.au

**Data Availability Statement:** The dataset used in this work is available from https://doi.org/10.1098/rsif.2013.1112 [Holloway PM, Angelova M, Lombardo S, St Clair Gibson A, Lee D, Ellis J. Complexity analysis of sleep and alterations with

## Abstract

### Background

Sleep disorders, such as insomnia, have been associated with extended periods of inactive, sedentary behaviour. Many factors contribute to insomnia, including stress, irregular sleep patterns, mental health issues, inadequate sleeping schedules, diseases, neurological disorders and prescription medications.

### Objectives

Identification of the patterns of sedentary time and its duration in adults with acute insomnia and healthy controls to determine the statistically significant sedentary bouts; comparison of the sedentary behaviour patterns in acute insomnia adults with healthy controls.

### Methods

We investigate the daytime actigraphy data and identify temporal patterns of inactivity among adults with acute insomnia and healthy adults. Seven days of actigraphy data were utilised to calculate sedentary time and bouts of variable duration based on a threshold of activity counts (<100 counts per minute). Statistical analysis was applied to investigate sedentary bouts and total sedentary time during weekdays and weekend. A logistic regression model has been used to determine the significance of sedentary bouts.

### Results

We found that individuals with acute insomnia accumulate a significant amount of their sedentary time in medium (6—30 minutes and 31—60 minutes) and longer (more than 60 minutes) duration bouts in comparison to healthy adults. Furthermore, a low $p$ value for total sedentary time ($2.54 \cdot 10^{-4}$) association with acute insomnia supports the finding that acute insomnia individuals are significantly more engaged in sedentary activities compared to healthy controls. Also, as shown by the weekend vs weekday analysis, the physical and sedentary activity patterns of acute insomnia adults demonstrate higher variability during the weekdays in comparison to the weekend.

insomnia based on non-invasive techniques. Journal of The Royal Society Interface. 2014;11 (93):20131112]. The authors confirm the data set is publicly available and that they did not have any special access privileges.

**Funding:** The author(s) received no specific funding for this work.

**Competing interests:** The authors have declared that no competing interests exist.

## Conclusion

The results of the present study demonstrate that adults with acute insomnia spend more time in low-intensity daily physical activities compared to healthy adults.

## Introduction

Our daily routines have changed drastically over the last few decades. Technology, social trends, and environmental factors have all contributed to sedentary behaviour, leading to significant amounts of sedentary time in our daily routine. As an example, objective monitoring indicates that people in the United States spend between seven and nine hours in a sedentary state during working hours [1]. A sedentary lifestyle can be defined as "any waking activity that requires ≤1.5 metabolic equivalents of effort while resting or reclining" [2]. As such, this term is used to describe activities such as sitting, reclining, watching television, reading, using a computer, and driving a vehicle. An equivalent metabolic rate represents the amount of oxygen consumed when sitting at rest [3]. A significant amount of research has been devoted to active lifestyles, physical activities and their effect on physical and mental health [4–8]. It has helped to advance our understanding of how regular exercise impacts health. Such studies, however, cannot provide conclusive evidence about the detrimental effects of sedentary behaviour on health because their design does not focus on inactivity [9]. In the last decade, researchers have increasingly focused on sedentary or inactive lifestyles in order to examine various aspects of these behaviours and on analysing patterns of sedentary time building up instead of simply estimating total sedentary time throughout the day [10].

There are several studies that have shown that physical activity has positive effects on mental and sleep health, including insomnia, sleep disorders, and depression symptoms [4, 5, 7]. Studies have also revealed that sedentary behaviour contributes to obesity [11], depression [12], cancer, sarcopenia, chronic obstructive pulmonary disease [13], physical ability, gait changes, Parkinson's disease, frailty and insomnia [14].

While sedentary behaviour cannot be defined as less physically active behaviour [15], only limited research has been conducted to study the effects of sedentary behaviour on sleep disorders [5–7]. Furthermore, sedentary behaviour is linked to depression and mental health problems [16], which can negatively impact sleep [17]. However, sleep problems such as sleep disorders and poor sleep quality have not yet been definitively linked to sedentary behaviour [18].

There are many people who suffer from insomnia, whether on a temporary basis, frequently, or chronically. Sleep may be adversely affected if one has difficulty falling asleep and staying asleep. This type of episode can last for a single short period or multiple short periods, appear at random times during the night or last long. People who have insomnia are often unable to sleep due to several factors, such as stress, illness, or pain. DSM-5 (Diagnostic and Statistical Manual of Mental Disorders, edition 5), defines insomnia as a sleep disorder characterized by persistent difficulty settling down, maintaining or returning to sleep. Insomnia can be divided into two main categories based on the duration of the sleep problems: acute insomnia, characterized by short-term sleep problems lasting for a few days to a less than three months; chronic insomnia, characterized by continuous sleep problems spanning at least three months.

Recently, the phenomenon of reduced and disturbed sleep has been recognized as a significant social and health issue which has an impact on the economy [19] and negatively affects health. In addition to contributing to a variety of health issues, this phenomenon has an adverse effect on quality of life [20]. The quality and quantity of sleep can be affected by

various types of sleep disturbances, of which insomnia is by far the most common. Most often, alterations in sleep patterns occur after a considerable amount of time, and as such, long-term monitoring may be necessary to detect any changes.

Studies addressing sedentary behaviour in association with insomnia mostly explore its association with physical activities as well as its potential role in alleviating insomnia symptoms [7, 21, 22]. In order to reduce the incidence of insomnia, it is strongly recommended that physical activity replaces sedentary behaviour.

Several studies concluded that excessive screen usage is positively related to shortened sleep time, sleep deprivation, and poor sleep quality, as well as insomnia [14, 23–28]. In addition to the considerable amount of work being undertaken to develop methods to evaluate physical activity and its positive health effects, very little has been done to evaluate sedentary behaviour parameters (sedentary time and breaks) from accelerometer data—particularly in relation to insomnia [29].

There are several methods for measuring sedentary behaviour such as self-report questionnaires, direct observation, diaries or logs, and activity trackers. During specific periods, self-report questionnaires are used to estimate the amount of time participants spend sitting or engaged in sedentary activities, whereas direct observation involves trained observers watching individuals closely and recording their sedentary behaviour. Another method is to ask participants to keep a diary or log where they can record the start and end times of their sedentary periods and any relevant information. The use of activity trackers such as pedometers or wearable devices can provide objective data regarding sedentary behaviour. In addition to measuring movement, these devices can also estimate the amount of time spent sedentary based on periods in which a person is not active. It is important to note that each screening method has its strengths and limitations, and the selection should depend on the specific research or practical purpose. In order to provide a more comprehensive assessment of sedentary behaviour, it is beneficial to integrate multiple methods, or to incorporate objective measures along with self-report measures.

Actigraphy is a relatively inexpensive and non-invasive measurement of human locomotor activity over an extended period of time using devices comparable by size to wristwatches. Through the use of actigraphy, it is possible to calculate sleep features based on physical activity [30–32] and this method is widely utilized to measure sleep and physical activity patterns over time in a home environment. Actigraphy devices are able to record in various time intervals such as 5, 10, 15, 30 and 60 seconds. The actigraphy devices allowed researchers to measure and analyze levels of physical activity ranging from purely sedentary to extremely vigorous in free-living settings over a period of time.

Motivated by the use of actigraphy in determining various activity patterns and our previous success in applying statistical and machine learning for the classification of insomnia based on actigraphic signals [31–34], we explore the daily sedentary behaviour patterns of individuals with acute insomnia and their healthy counterparts.

The main aim of this study is to investigate the differences in sedentary behaviour in individuals with acute insomnia and age-matched healthy controls using actigraphy data.

Our hypothesis is: *Adults with acute insomnia spend more time sedentary than healthy control subjects.*

In order to prove or disprove this hypothesis, the first objective of this work is to investigate the patterns of sedentary time spent by adults with acute insomnia and age-matched healthy controls. Sedentary time has been measured with actigraphy most commonly by stratifying it into 1-minute, 5-minute, 10-minute, 20-minute, 30-minute, or 60-minute intervals [10]. However, no clinically approved rules or protocol for the significance of the duration of sedentary bouts have been demonstrated in the literature. Thus, the second objective of the study is to

determine the statistical significance of the duration of various sedentary bouts for adults with acute insomnia and compare it with age-matched healthy controls.

Based on the statistical analysis of the patterns of daytime activity, the results of this study could provide new insights into the accumulation of sedentary time in adults with acute insomnia. The study can further be used as a basis for developing methods for reducing inactive behaviour in individuals with insomnia. In addition, it can be used to determine the effects of sedentary behaviour on health and well-being. To our knowledge, this is the first study to examine patterns of sedentary behaviour using daytime actigraphy data obtained from individuals with acute insomnia and compare them with age-matched healthy controls.

This paper is organized as follows. Materials and methods section is devoted to the description of the data, descriptive statistics, and methods to compute sedentary and physical activities as well as the logistic regression model. In the Results section, analysis of the sedentary and physical activity patterns is provided and further discussed in Discussion section, where our hypothesis is verified. In Conclusions, we summarise our findings in relation to the objectives of this study.

## Materials and methods

### Data collection and pre-processing

A publicly available actigraphic dataset obtained from adults suffering from acute insomnia and healthy control group was used in this study [34] (supplementary material). The data collection for the original study was approved by the University of Glasgow Ethics Committee. Informed written consent was obtained from all participants during the original data collection. The dataset contains actigraphic data for 37 adults (age: 19-40 years) including 15 individuals with acute insomnia (AI) and 22 healthy controls (HC).The study initially intended to include age-matched participants ranging from 20 to 60 years of age. However, there were no participants in the age groups of 50 to 60 in the healthy control group data set. Therefore, it was necessary to select individuals with ages between 20 to 40 years old to avoid imbalance between the groups. Fig 1 presents the data inclusion and exclusion criteria of this study.

Data were collected using the Actiwatch monitor (AW4) from Cambridge Neurotechnology. It is lightweight, wearable wristwatch-like device that detects maximum amplitude of the physical motion accelerations and converts them into activity counts. These counts represent the peak acceleration detected over each epoch [35]. "Epochs" are intervals of time when accelerometers integrate activity movements [36]. For this study, seven days of daytime actigraphy data (between 8am and 10pm) were used, which were integrated over one-minute intervals.

To deal with the missing data (approximately 1% per actigraphy series), we used the moving median method over a shifting window of 30 minutes. The data points are calculated from local $m$-point medians [37]. Each median is calculated across neighbouring elements of the time series $X$ within a sliding window of length $m$. We performed all analyses using Matlab Statistics and Machine Learning Toolbox (Matlab version: 9.12.0.1884302 (R2022a)).

### Statistical methods

Statistical analysis of seven days of raw actigraphy daily data was conducted in order to gain a more comprehensive understanding of the sedentary behaviour in the cohort. The mean and standard deviation were calculated from the daytime activity data of AI adults as well as HC adults of the same age range (between 20 and 40 years).

A variety of sedentary bout duration bins were introduced. The total time spent in each of the duration ranges and the counts for each of the bout duration bins were calculated in order

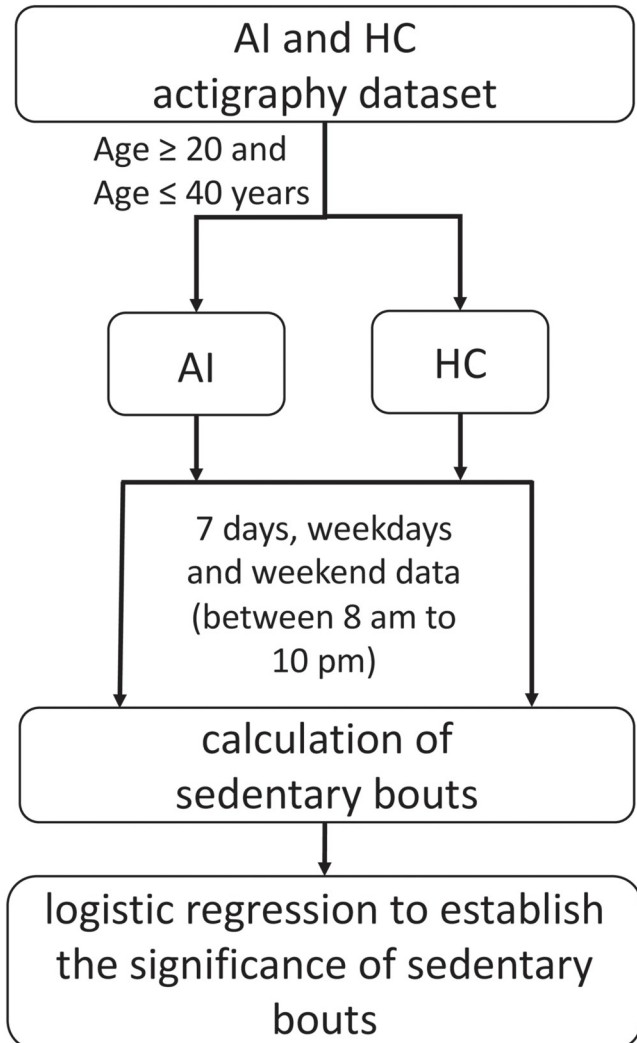

**Fig 1. Process flowchart including data inclusion/exclusion criteria for individuals with acute insomnia (AI) and the healthy controls (HC).**

to analyze how individuals accumulate their inactivity time on the daily basis. As sedentary bouts over time do not follow a normal distribution, we used an exponential distribution to analyze sedentary bouts acquired over time.

**Exponential distribution.** Exponential distribution is a one of the most common continuous distributions that describes the time difference between independent random events. The exponential distribution is often used to predict the timing between Poisson point process events [38] which take place indiscriminately at a regular average rate. The exponential distribution is a memoryless distribution and it is possible for the exponential random variable to have either more small values or fewer larger values.

As going into a sedentary bout and going out of a sedentary bout are random and independent events, it would be natural to expect the time between these events to be distributed exponentially. Furthermore, sleep bouts have been experimentally confirmed to have exponential distribution [39].

The exponential distribution is uniparametric and can be defined as follows:

$$f_{X|\lambda}(x) = \left\{ \begin{array}{cc} \lambda \exp^{-\lambda x}, & x > 0 \\ 0, & x \le 0 \end{array} \right\}, \tag{1}$$

where $X$ is the random variable that is distributed according to the probability density function $f$ anx $\lambda$ is the rate parameter. Notably, for the exponential distribution the mean is equal to the standard deviation, $\mu = \sigma = 1/\lambda$.

**Two-sample Kolmogorov-Smirnov test ($K - S$).** The two-sample Kolmogorov-Smirnov (K-S) test establishes whether the two samples belong to the same distribution. This is a nonparametric test and a very useful method for assessing models that are based on regression and classification. This test relies on the empirical cumulative distribution function (ECDF) for a random data $X = x_1, \ldots, x_N$:

$$F_X(x) = \frac{1}{N} \sum_{i=1\ldots N} 1_{x_i \le x}, \tag{2}$$

where $N$ is the number of independent and identically distributed ordered observations $X$. The two-sample K-S test coefficient $D$ estimates the most significant vertical distance between the ECDFs $F_X$ and $F_Y$ for two random variables $X$ and $Y$, calculated using Eq 2,

$$D = \sup_x |F_X(x) - F_Y(x)|, \tag{3}$$

where $\sup_x$ is the supremum of the set of distances. The $p$ value can be obtained from $D$ using, for example, the original method [40], or readily available tables in [41] and accounting for the sample size with $z = D\sqrt{n}$. When the calculated $p$ value is smaller than some arbitrarily chosen small value (in this study we accept $p$ value threshold of 0.05 significance levels), the null hypothesis is ruled out. We use the Matlab implementation of the K-S test.

## Sedentary activities measurement

The term "sedentary activity" refers to any activity not exceeding 100 activity counts per minute measured by an actigraphy device [42–44]. The majority of studies employ the arbitrary and universally accepted activity counts cutoff threshold recommended by Freedson [45], which is less than 100 activity counts per minute for determining sedentary behaviour. In this paper, we follow this recommendation and consider a cutoff <100 counts per minute (cpm) to classify the activity as sedentary behaviour [45].

The following sedentary parameters were determined for each day for each person and averaged over the seven days over the number of individuals in each group (AI and HC groups).

- sedentary bout (`sed_bouts`)—a continuous period of sedentary time during which the activity count is less than 100 cpm.

- total sedentary time (`sed_total`)—provides the total amount of time spent in inactivity during the day; the entire time in which activity counts are less than 100 cpm.

If only the total sedentary time were considered, there would be no clarity of when and how sedentary time accumulated during the day, as it only provides the total duration of several inactivity periods. In our study, it is necessary to determine sedentary behaviour in bouts with various length. We calculated the amount of time spent in sedentary bouts (`sed_bouts`) of short (1—3 minutes and 4—5 minutes), medium (6—30 minutes, 31—60 minutes), and long

duration (more than 60 minutes). Furthermore, we calculated the total sedentary time, which is the sum of all the counts between 8 am and 10 pm. All sedentary activity measurements obtained for each day (8 am to 10 pm) were then averaged over seven days over the number of participants in each group (AI and HC) in order to determine the average sedentary values.

## Physical activities measurement

Physical activity (PA) is a challenging and intermittent behaviour, which is not straightforward to quantify due to its large variability. A strong association exists between PA and a healthy lifestyle. As well as reducing rates of all-cause mortality, PA can lower the risk of colon cancer, cardiovascular disease, hypertension and type 2 diabetes [46]. Actigraphy devices measure acceleration and provide activity counts as their output metric. In order to categorize activities by their intensity, such as light, moderate, or vigorous, researchers use thresholds to differentiate activities. Rather than analyzing low, moderate and vigorous PAs separately, this study looks at the PAs where the activity count exceeded 100 cpm collectively. We have calculated total activity time as the sum of all counts above 100 cpm between 8 am to 10 pm.

We introduce `wake_total` parameter, which is the total physical activity time when activity count is $\geq$100 counts per minute (cpm). This parameter was averaged over seven days and over the number of individuals in both AI and HC cohort.

## Logistic regression

Logistic regression is a supervised statistical learning and classification algorithm that performs prediction using the concept of probability. It falls under the generalized linear models (GLMs) class [47]. Although logistic regression uses predictive modelling as a regression, it is still a binary classification algorithm. In this method, a set of independent variables is used to predict the categorical dependent variable.

As part of the logistic regression, a logit transformation is applied to the odds, which is defined as the likelihood of success divided by the likelihood of failure. Logistic regression can be described by the following equation:

$$\ln\left(\frac{p}{1-p}\right) = \beta_0 + \beta_1 x_1 + \beta_2 x_2 + \ldots + \beta_n x_n, \tag{4}$$

where an event's probability is represented by $p$, the $y$-intercept or bias is denoted by $\beta_0$, and the independent variables are represented by $x_1 \ldots x_n$. The coefficients $\beta_1$ to $\beta_n$ describe the association between each independent variable and the outcome (log odds) [48]. In logistic regression:

(i) Estimated coefficients $\beta_i$ reflect the deviation in the mean of the dependent variable when the independent variable is shifted by one unit without changing the other variables in the model. In order to evaluate each variable independently, it is imperative that all the other variables must be kept constant.

(ii) The standard error (SE) of estimate indicates the average difference between the observed value and the regression curve. It allows calculating the error rate of the regression model.

(iii) $p$ value represents the probability of getting a coefficient value of at least the same magnitude as the estimated coefficient assuming null hypothesis. Lower probabilities provide substantial evidence that refutes the null hypothesis. The significance level is set in order to determine whether a study is statistically significant. There is enough evidence to reject the

null hypothesis when the *p* value is less than the set significance level. As a rule of thumb, many studies use *p* value <0.05 as statistically significant [49].

Logistic regression can provide predictions as well as probabilities of events that will occur. This makes it a popular approach for classification because of the ease of interpretation. Furthermore, it also measures the significance of each dependent variable's contribution with the extent of its reliability [50]. In this study, a logistic regression procedure is used to analyze the statistical significance of the sedentary behaviour variables (`sed_bouts`).

## Results

Table 1 illustrates the total number of participants aged between 20 and 40, male and female numbers, age (mean and median) and the total number of recording days for the age-balanced dataset used in this study.

Sedentary periods with five different lengths (`sed_bouts`), `sed_total` and `wake_total` were calculated using actigraphy measurements in adults with AI and healthy adults in order to analyze the patterns of sedentary behaviour. The results were statistically analyzed and their statistical significance was assessed using the logistic regression model.

### Results of statistical analysis

Table 2 illustrates the statistical analysis of raw signal data for individuals averaged over the seven days period and aggregated over the number of participants in the AI and HC groups. The overall mean value of the activity signal data for the AI group is 134.57 which is substantially lower than the mean value of the HC group, 184.51. The overall average standard deviation (sd) value of the daytime actigraphy of AI group 197.20 is also lower than that of the HC group 224.72.

Based on the exponential distribution used to fit the data, Table 3 shows the mean values with lower and upper bounds ranges, of the total amount of time spent in short (1 minute—3 minutes and 4 minutes—5 minutes), medium (6 minutes—30 minutes and 31 minutes—60 minutes) and long (above 60 minutes) duration sedentary bouts, total sedentary time (`sed_total`) and total activity time (`wake_total`) averaged over the seven days and over the number of individuals of both cohorts (AI and HC). In addition, a *p* value computed using two samples K-S test is also provided in Table 3. The lower bound refers to the lowest permissible value, and the upper bound refers to the highest acceptable value. Compared to HCs, individuals with AI have high lower and upper confidence values of all different sedentary bout's average values. The upper bound confidence value (93.06) of the mean for `sed_bouts` (>60 minutes) of the HC cohort is lower than AI's lower bound confidence value (102.64). The AI group differs from the HC group not only in terms of mean values but also in terms of the ranges of lower and upper bounds.

**Table 1. Total number of subjects, male and female, average age and the total number of days of actigraphy recordings for the subjects used in this study.**

|  | Adults with acute insomnia (AI) | Healthy controls (HC) |
| --- | --- | --- |
| Number of Subjects | 15 | 22 |
| Gender | 3 males, 8 females and 4 unknown | 8 males and 14 females |
| Mean age (years) | 28.87 | 27.82 |
| Median age (years) | 26 | 25 |
| number of days | 105 | 154 |

**Table 2. The mean and standard deviation (sd) of the raw activity signals (counts per minutes, cpm) over the seven days for the AI (adults with acute insomnia) and HC (healthy control) cohorts.**

|  | AI (adults with acute insomnia) | HC (healthy control) | $p$ (K-S) |
|---|---|---|---|
| mean | 134.57 ± 54.08 | 184.51 ± 106.84 | 0.09 |
| sd | 197.20 ± 72.73 | 224.72 ± 85.67 | 0.75 |
| mean (cpm <100) | 19.32 ± 5.93 | 23.49 ± 5.12 | 0.10 |
| sd (cpm <100) | 25.95 ± 4.39 | 28.33 ± 2.43 | 0.20 |
| mean (cpm ≥100) | 332.90 ± 83.87 | 354.92 ± 96.62 | 0.75 |
| sd (cpm ≥100) | 222.38 ± 79 | 234.93 ± 81.99 | 0.51 |

## Patterns of sedentary activities

We estimated the total time spent in inactivity by adding up minutes for occurrences within established cut points of activity counts (<100 cpm). Nevertheless, it is important to analyze how the time was accumulated, with a particular focus on the time of day and duration of these inactive periods. Additionally, we looked at the weekday and weekend analyses to determine how this pattern differs. The results of our analysis for the sedentary activities for seven days, weekdays and weekends are shown in Tables 3 and 4.

**Sedentary behaviour (7-days total).** Table 3 presents the outcome of sedentary activities averaging over seven days and over the participants in the AI and HC groups. Averaged over the number of individuals in the AI and HC groups for seven days, the average total amount of sedentary time per individual of the AI cohort is 541.65 minutes from 8 am to 10 pm, which is high compared to healthy controls (460.94 minutes). A very low $p$ value from K-S test ($1.8 \cdot 10^{-4}$) supports the hypothesis that people with acute insomnia tend to be less active on average during the day than people in the healthy control group (HC). Thus, Table 3 shows that individuals with acute insomnia are less active compared to age-matched healthy adults.

HC cohort has spent more time in short-term sedentary bouts (1—3 minutes and 4—5 minutes). However, AI adults were more involved in sedentary activities for longer duration accumulating sedentary time in medium (6—30 minutes, 31—60 minutes) and long duration bouts (>60 minutes).

HC cohort spent on average 94.27 minutes over the time period of seven days in a 1—3 minutes sedentary periods. The AI group spent less time in this sedentary period range (81.80 minutes).

**Table 3. Mean ± lower/upper bound values of 95% confidence interval for short, medium and long sedentary bouts duration, `sed_total` (total sedentary time) and total physical activity time (`wake total`) using seven days of actigraphy data, averaged over the individuals in each group (AI and HC).** The upper bound and lower bound confidence values provide the range of the calculated mean. $p$ value is calculated using K-S test.

| | AI (adults with acute insomnia) | HC (healthy control) | |
|---|---|---|---|
| Sedentary bouts | mean (minutes) | mean (minutes) | $p$ (K-S) |
| `sed_bouts` (1-3 minutes) | $81.8^{+100.01}_{-68.16}$ | $94.27^{+111.12}_{-80.99}$ | 0.01 |
| `sed_bouts` (4-5 minutes) | $40.74^{+49.81}_{-33.95}$ | $44.45^{+52.40}_{-38.19}$ | 0.58 |
| `sed_bouts` (6-30 minutes) | $218.45^{+267.08}_{-182.02}$ | $187.23^{+220.71}_{-160.85}$ | 0.04 |
| `sed_bouts` (31-60 minutes) | $77.48^{+94.73}_{-64.56}$ | $56.05^{+66.07}_{-48.15}$ | 0.002 |
| `sed_bouts` (>60 minutes) | $123.18^{+150.61}_{-102.64}$ | $78.94^{+93.06}_{-67.82}$ | 0.74 |
| `sed_total` | $541.65^{+662.24}_{-451.33}$ | $460.94^{+543.36}_{-395.99}$ | 1.8E-04 |
| `wake_total` | $297.51^{+363.74}_{-247.89}$ | $378.39^{+446.06}_{-325.08}$ | 1.8E-04 |

**Table 4. Mean ± lower/upper bound values of 95% confidence interval for five different bins of sedentary bouts duration, total sedentary time, and total physical activity time averaged over five days during the week and two days during the weekend for the AI (adults with acute insomnia) and HC (healthy control) group.** *p* values are obtained using K-S test.

| Sedentary bouts | AI (adults with acute of insomnia) mean (minutes) | HC (healthy control) mean (minutes) | *p* (K-S) |
|---|---|---|---|
| | Weekday | | |
| sed_bouts(1-3 minutes) | $83.32^{+105.93}_{-67.27}$ | $95.04^{+115.63}_{-79.51}$ | 0.01 |
| sed_bouts(4-5 minutes) | $41.25^{+52.45}_{-33.30}$ | $45.15^{+54.81}_{-37.68}$ | 0.76 |
| sed_bouts(6-30 minutes) | $228.24^{+290.17}_{-184.26}$ | $188.46^{+229.30}_{-157.66}$ | 0.01 |
| sed_bouts(31-60 minutes) | $72.49^{+92.16}_{-58.53}$ | $48.90^{+59.50}_{-40.91}$ | 0.01 |
| sed_bouts(>60 minutes) | $120.41^{+153.09}_{-97.21}$ | $81.53^{+99.20}_{-68.20}$ | 0.86 |
| sed_total | $545.72^{+693.80}_{-440.57}$ | $458.96^{+558.43}_{-383.96}$ | 0.0023 |
| wake_total | $293.427^{+373.05}_{-236.89}$ | $380.43^{+462.88}_{-318.26}$ | 0.0023 |
| | Weekend | | |
| sed_bouts(1-3 minutes) | $78^{+115.61}_{-56.18}$ | $92.34^{+127.09}_{-70.15}$ | 0.44 |
| sed_bouts(4-5 minutes) | $39.47^{+58.50}_{-28.43}$ | $42.98^{+59.15}_{-32.65}$ | 0.28 |
| sed_bouts(6-30 minutes) | $193.97^{+287.49}_{-139.72}$ | $184.16^{+253.45}_{-139.90}$ | 0.35 |
| sed_bouts(31-60 minutes) | $89.93^{+133.30}_{-64.78}$ | $73.91^{+101.72}_{-56.15}$ | 0.24 |
| sed_bouts(>60 minutes) | $130.10^{+192.83}_{-93.71}$ | $72.48^{+99.75}_{-55.06}$ | 0.33 |
| sed_total | $531.47^{+787.71}_{-382.82}$ | $465.86^{+641.15}_{-353.90}$ | 0.0666 |
| wake_total | $307.70^{+456.06}_{-221.64}$ | $373.30^{+513.76}_{-283.58}$ | 0.0666 |

The AI group had accumulated a high average time in inactive behaviour in long-duration bouts; 218.45, 77.48 and 123.18 minutes in 6 minutes—30 minutes, 31 minutes—60 minutes and >60 minutes sedentary bouts respectively with low *p* values of K-S tests. The AI group also had a higher mean value with a higher lower and upper-bound confidence values in long-duration bouts than the HC group, which indicates that the AI group is generally involved in low-intensity activities during the day. Thus, medium and longer-duration sedentary bouts are significant for adults with acute insomnia.

Fig 2 shows AI and HC cohorts two-hour averages of the time spent in sedentary activities during the day. Specifically, this depicts times during the day when people with acute insomnia are more inactive than healthy controls. Fig 2 clearly illustrates the heaviest sedentary periods of AI adults occur throughout the day and from 6 pm to 10 pm at night the average of activity count is approximately the same. As compared to healthy controls, individuals with acute insomnia spend a greater amount of time passively during the day. Fig 3 shows the total amount of sedentary time during 2 hours (cpm<100). It clearly shows that adults with acute accumulates more time in sedentary activities compared to healthy adults.

**Sedentary behaviour (weekdays and weekends).** The weekday and weekend sedentary values for the AI and HC groups are shown in Table 4: the five-day aggregated values for weekdays and two-day aggregated averages for weekends, of short (1—3 minutes and 4—5 minutes), medium (6—30 minutes and 31—60 minutes) and long (above 60 minutes) duration sedentary bouts, total sedentary time (sed_total) and total activity time (wake_total) averaged over the participants of both control groups. AI participants' weekend sedentary activity patterns are similar to their weekday inactivity patterns, although time accumulated in sedentary bouts is less distinct on weekends.

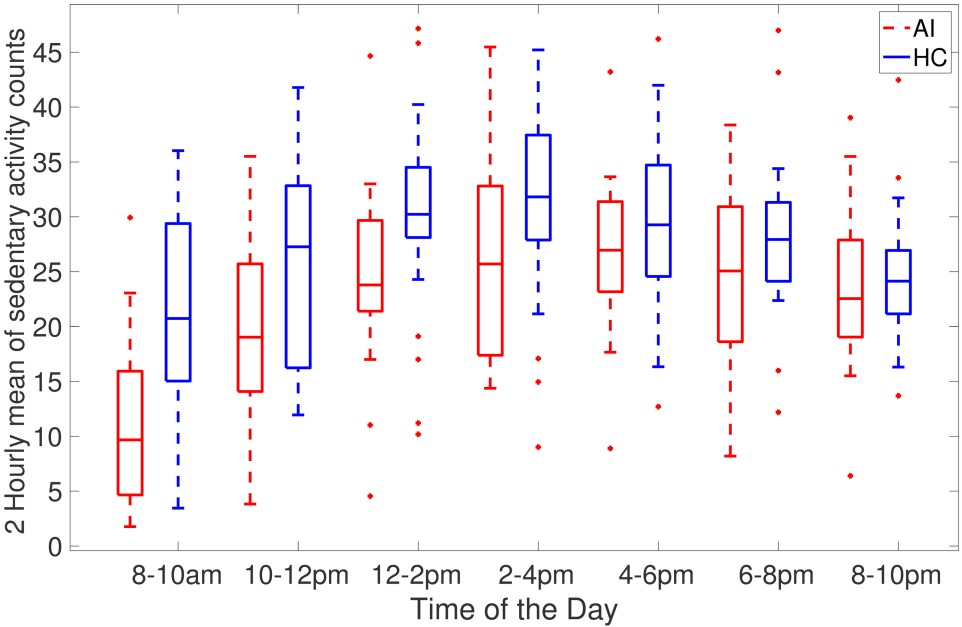

**Fig 2. Box plots of 2-hour mean of sedentary activity counts (where activity counts are less than 100 per minute) averaged across 7 days for AI (red) and HC (blue) individuals.**

On average, the total time (`sed_total`) spent in sedentary activities is higher in the AI group (545.72 minutes) as compared to the HC group (458.96 minutes) on weekdays. A similar pattern is observed on weekends: AI spend 531.47 minutes and HC spend 465.86 minutes on average in an inactive lifestyle.

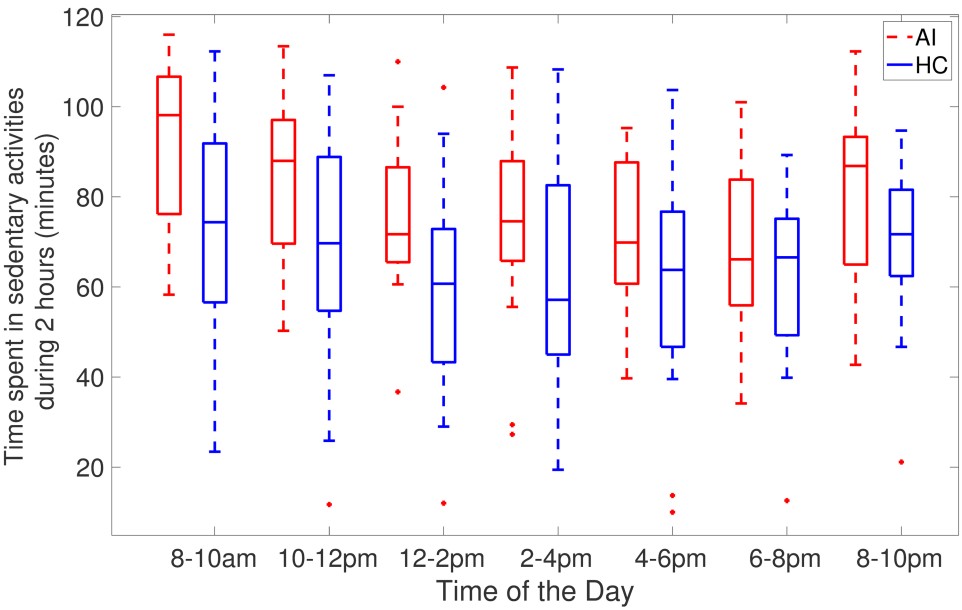

**Fig 3. Box plots of total time spent in sedentary activity 2 hourly across 7 days for AI (red) and HC (blue) individuals where activity counts are less than 100 counts per minute.**

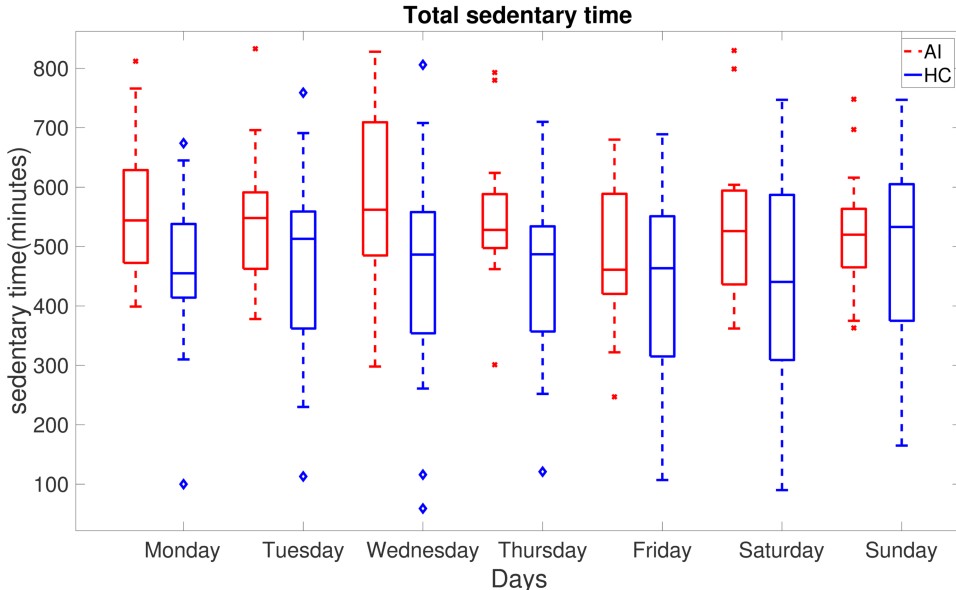

**Fig 4. Boxplots of total sedentary time day-wise averaged over the participants in AI (red) and HC (blue) groups.**

Fig 4 illustrates the averaged total sedentary time on day by day basis for AI (red) and HC (blue) group members. The overall average of total sedentary time (averaged across each group) is consistently high for individuals suffering from acute insomnia on weekdays. In contrast, the average amount of total sedentary time for both groups is similar on weekends.

The HC group accrues a greater amount of sedentary time in shorter periods (1 minute—3 minutes and 4 minutes—5 minutes) regardless of whether it is a weekday or a weekend. Meanwhile, the AI group spends more inactive time in medium (6 minutes—30 minutes and 31 minutes—60 minutes) and longer (>60 minutes) bouts. The AI group accumulated almost 1.5 times more inactive time in long duration (>60) bouts regardless of weekdays or weekends. Additionally, the significant low $p$ value of sedentary bouts of 6 minutes—30 minutes and 31 minutes—60 minutes length, and total sedentary values from the K-S tests (0.1, 0.01 and 0.00230) indicate that adults with acute insomnia are more involved in passive behaviour than healthy adults during weekdays. The difference between aggregated time in sedentary bouts and total sedentary time between AI and HC is smaller during weekends, although AI remains more sedentary than HC. The K-S test $p$ values are not very low, suggesting that most people lead sedentary lifestyles on weekends.

Fig 5 illustrates how time spent in sedentary activities was distributed over the 7 days, separately into medium (6—30 minutes and 31—60 minutes) and longer (> 60 minutes) duration episodes. It can be seen from all the box plots that the accumulated sedentary time in the AI group is more varied and different compared to the HC group.

## Patterns of physical activities

This study mainly focuses on sedentary activities, nevertheless we present the results for physical activities in Tables 3 and 4 which provide the average total active time from 8 am to 10 pm over 7 days, weekdays and weekends, given the activity counts are greater than 100 per minute. The average of `wake_total` for the AI group is 297.51 minutes which is lower than the HC group, 378.39 minutes. The high average physical activity value supports that the HC cohort

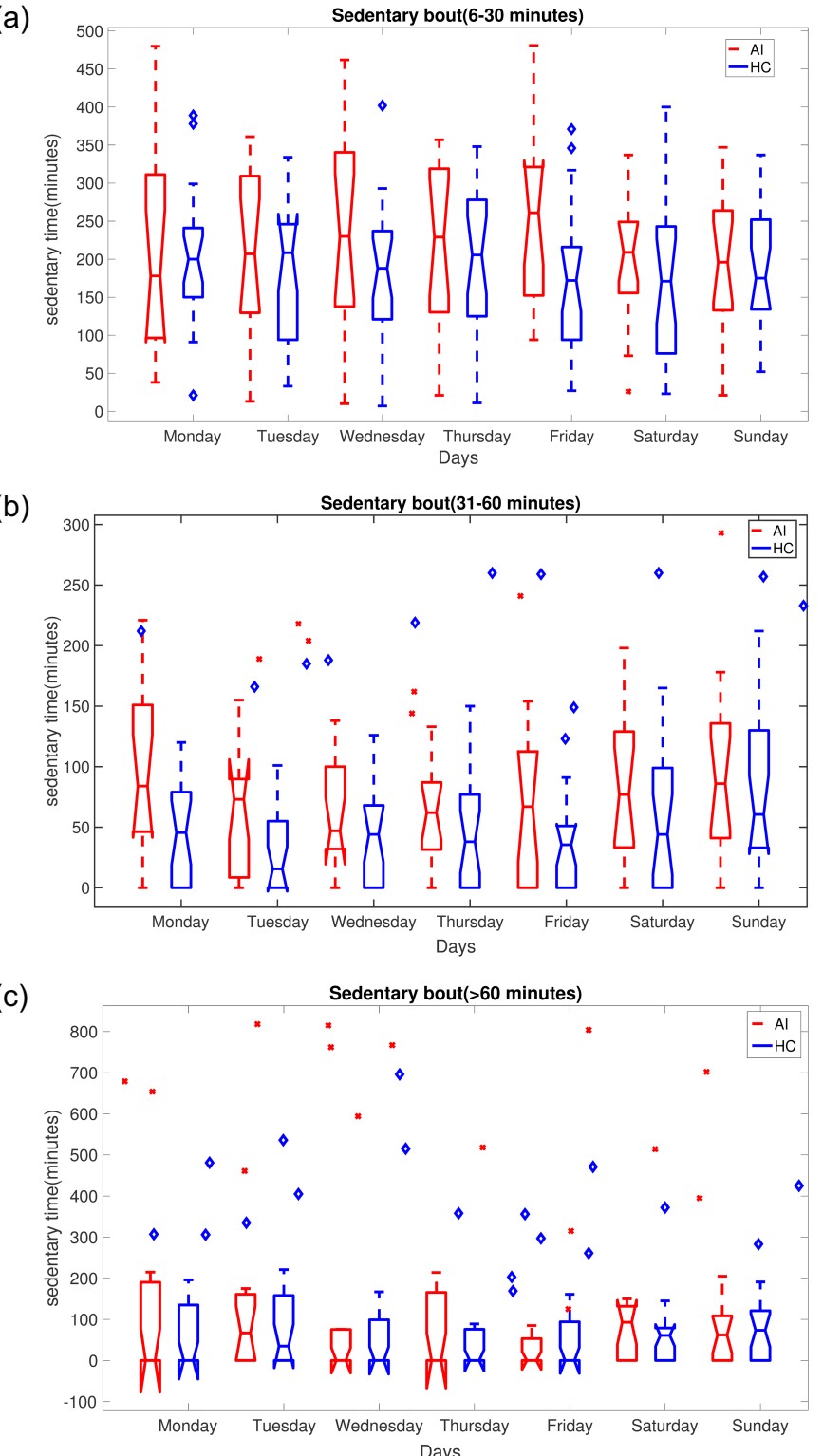

**Fig 5. Boxplots of average time spent in sedentary bouts of different duration on a day-by-day basis by AI (red) and HC (blue) cohorts.** Sedentary bouts are shown for: top—96-30 minutes; middle—31-60 minute; bottom—>60 minutes.

spends significantly more time participating in strenuous physical activities than the AI group. The range of upper and lower bounds of mean `wake_total` of the AI group illustrates that individuals with acute insomnia are involved in less energetic physically intensive activities. Analysis of the weekday and weekend physical activities separately for AI and HC groups shows a similar pattern.

An illustration of the time spent in physical activities within 2 hours where participants were physically active ($\geq$100 cpm) between 8 am and 10 pm is provided in Fig 7, which shows that the individuals with acute insomnia on average are less active or engage in less strenuous activities throughout the day than those with HC. This figure represents the average of activity counts over two hours (where cpm> = 100), indicating that even though the AI group is physically active, their level of intensity is lower. It is interesting to note that the 2-hour activity times for AI and HC cohorts are similar only for the evening interval (6-8 pm). Also, the peak activity hours are different for AI (6-8 pm) and HC (2-4 pm) cohorts, with the AI cohort activity uniformly increasing towards the late evening period and HC cohort activity increasing towards 2-4 pm interval and then decreasing towards the night. It can be further speculated that lack of sleep during the night for the AI cohort is partially compensated through sedentary activities during the day hours.

## Logistic regression

To further demonstrate statistically significant differences in sedentary behaviour between the healthy cohort and insomnia sufferers, as well as to analyse the relevance of the different sedentary behaviour duration bins to insomnia, we performed logistic regression on the data. The data was labelled based on subjects falling into one of two categories (one-hot-coded as 0 for acute insomnia and 1 for healthy subjects). The average duration for five different types of sedentary bouts for each of the 37 subjects represents an individual predictor variable, and no interaction between individual predictors was included in the model. Logistic regression coefficients $\beta_1 \ldots \beta_5$ indicate the significance of the particular sedentary bout duration bin on the target variable.

The coefficients and their statistical parameters are shown in Table 5. Note that according to the model definition, the negative $\beta_i$ coefficients indicate that the probability of a subject to be labelled as 1 (healthy) decreases with increasing duration of the sedentary bouts of a particular length. This behaviour is observed for all bins with sufficiently low (<0.05) $p$ values, indicating that the longer sedentary bouts are indeed characteristic for insomnia sufferers. On the other hand, the bins of shorter duration (1—3 minutes and 4—5 minutes) have high $p$ values, further supporting their statistical insignificance and low relevance to the behaviour of the insomnia individuals during daytime.

**Table 5. Logistic regression coefficients calculated for seven days average sedentary bouts data for AI (labelled as 0) and HC (labelled as 1) cohorts.** $\beta$ are calculated parameter estimates, and SE indicate standard errors for the parameter estimates. $p$ values are provided for the null hypothesis (absence of dependence, the corresponding coefficient is equal to zero).

| | Estimate($\beta$) | SE | $p$ |
|---|---|---|---|
| Intercept ($\beta_0$) | 2.242 | 0.861 | 0.009 |
| sed bouts1-3 ($\beta_1$) | -0.004 | 0.006 | 0.504 |
| sed bouts4-5 ($\beta_2$) | 0.006 | 0.008 | 0.483 |
| sed bouts6-30 ($\beta_3$) | -0.005 | 0.002 | 0.002 |
| sed bouts31-60 ($\beta_4$) | -0.005 | 0.002 | 0.031 |
| sed bouts>60($\beta_5$) | -0.003 | 0.001 | 0.012 |

Standard error estimates (SE) indicate how far a particular parameter in a data set is from the representative population. The observed very low SE values for longer bouts imply that the observed values are close to the regression line.

## Discussion

Insomnia and other sleep-related problems often result in fatigue and sleeping during the daytime. The inability to stay awake is often referred to as sleepiness, and fatigue refers to deteriorating cognitive and motor capabilities, as well as a decreasing motivation and desire to rest more [51]. In some cases, people suffering from sleep problems may be unwilling to engage in physical activity. It is possible that insufficient sleep and resulting tiredness can contribute to an increase in sedentary behaviour since individuals may feel less motivated or physically unable to engage in physical activity [52]. Additionally, insomnia can contribute to excessive daytime sleepiness, resulting in individuals turning to sedentary activities in an effort to alleviate fatigue [53]. Furthermore, individuals with acute insomnia may be likely to avoid stimulating or physically demanding activities for the purpose of facilitating relaxation and improving their chances of falling asleep [54].

Sedentary time is commonly estimated by measuring the duration of periods with activity counts less than the threshold for sedentary activities from actigraphy devices [55, 56]. However, this method clearly ignores the long-term characteristics of the definition of sedentary behaviour. The pivotal research by Healy et al. [55, 57] explained the intricacy of this issue which substantiated that interruption in sedentary time leads to disparities in cardio-metabolic and inflammatory risk biomarkers. We computed various lengths of sedentary bouts to examine sedentary behaviour in daily activities of adults with acute insomnia and healthy controls. Compared to adults with acute insomnia, healthy adults have a higher mean and standard deviation of recorded activity signals during the day, which suggests that they are more active throughout the day. The analysis, based on the exponential distribution of sedentary bouts, reveals that healthy controls have accumulated more time sedentary over 1—3 minutes and 4—5 minutes compared to individuals with acute insomnia. It was found that adults with acute insomnia accumulated more time in periods of inactivity between 6 and 30, 31 to 60, and more than 60 minutes.

Sedentary time accumulated over shorter periods of time (less than five minutes) is negatively correlated with health risk factors [29]. This pattern was inverted when the sedentary time was computed as prolonged periods of time when bouts lasted longer than 10 minutes [29]. Health-related issues might differ among individuals who engaged in the same level of sedentary activities, depending on how their sedentary activity was accumulated.

The main finding of this study is that individuals with acute insomnia spend more time sedentary throughout the day (Table 3) and engage in low-intensity physical activities more than their healthy counterparts. Fig 7 also shows that the average time spent in physical activities with activity counts greater than 100 (high-intensity physical activity) is higher for the healthy control group. Furthermore, our results demonstrate that the average physical activity is higher for the healthy control group than for the acute insomnia group (see Tables 3 and 4), although the difference may not be statistically significant.

Individuals may differ in their sedentary behaviour on weekdays and weekends depending on their personal circumstances and lifestyle factors such as work, school, and commuting needs during the week. The structured routine of the week may limit opportunities for physical activity and increase sedentary behaviour. Additionally, most people commute to work or school on weekdays. This often involves sitting for extended periods of time, whether driving, taking public transportation or working from home. As opposed to weekdays, weekends offer

more flexibility. So people tend to catch up on sleep, complete house chores and engage in social and recreational activities. The differences between weekdays and weekends are universal but they can vary based on personal preferences. Our results show that individuals with acute insomnia are more sedentary compared to the healthy controls during the weekdays, however, the difference between these cohorts in the sedentary bouts duration and total sedentary time during the weekends is smaller. On the other hand, healthy controls are more sedentary and less physically active during the weekend as compared to their behaviour on weekdays. This could also be due to the limitations of the collected data as we only have 2 days of the weekend data. Individuals with acute insomnia spend more time sedentary in comparison to healthy adults as indicated by `sed_total` and `sed bouts (>60)` in Tables 3–5. In addition to spending more time sedentary, individuals with acute insomnia spend more time in sedentary activities and have a lower intensity of activity counts (Figs 2, 3, 6 and 7) during the day. Our hypothesis, that adults with acute insomnia spend more time in lower energy expenditure activities compared with healthy individuals, is also supported by lower $p$ values ($<0.5$) of various sedentary bouts as well as total sedentary time. A logistic regression analysis was conducted using short, medium and longer-duration bouts estimated from 7 days of activity data in order to determine the significance of the associated sedentary bouts. There is a statistically significant impact of medium and longer duration bouts of sedentary behaviour on AI/HC label, according to lower $p$ values and negative estimates coefficients. Induced by a slight increase in sedentary activity episodes, the probability of acute insomnia can increase significantly.

These analyses, however, have several limitations. Firstly, it is difficult to compare the collected data results as the primary purpose of data collection used in this study was not focused on measuring sedentary behaviour and physical activity, but rather on studying sleep in adults

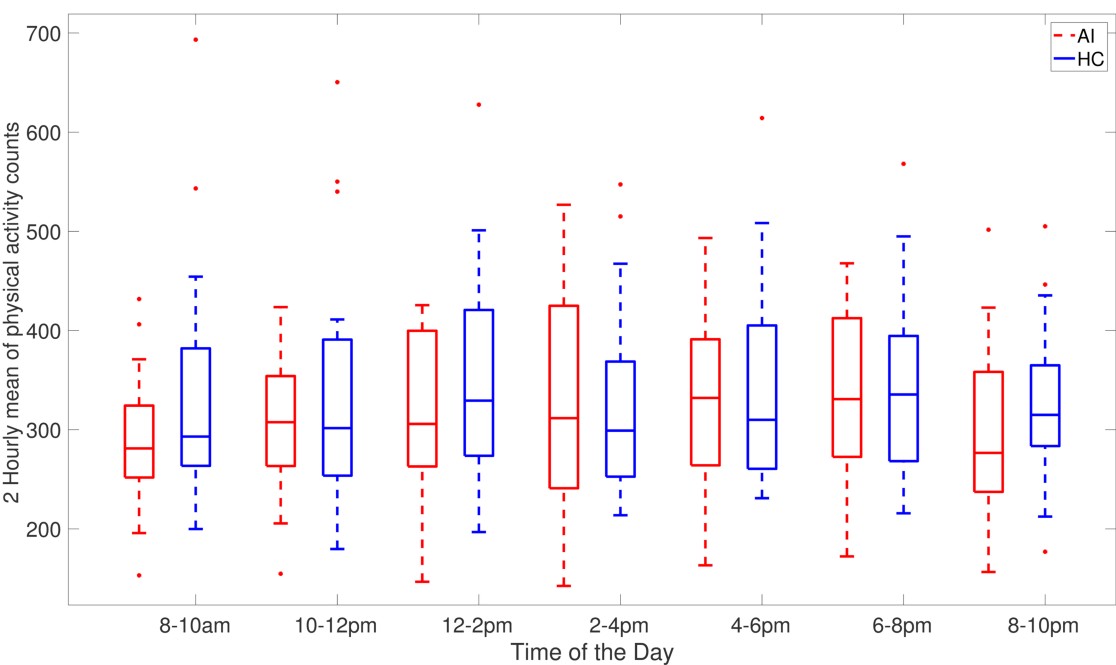

**Fig 6. The physical activities between 8 am to 10 pm averaged across seven days by the number of participants of each group where activity counts are greater than and equal 100.**

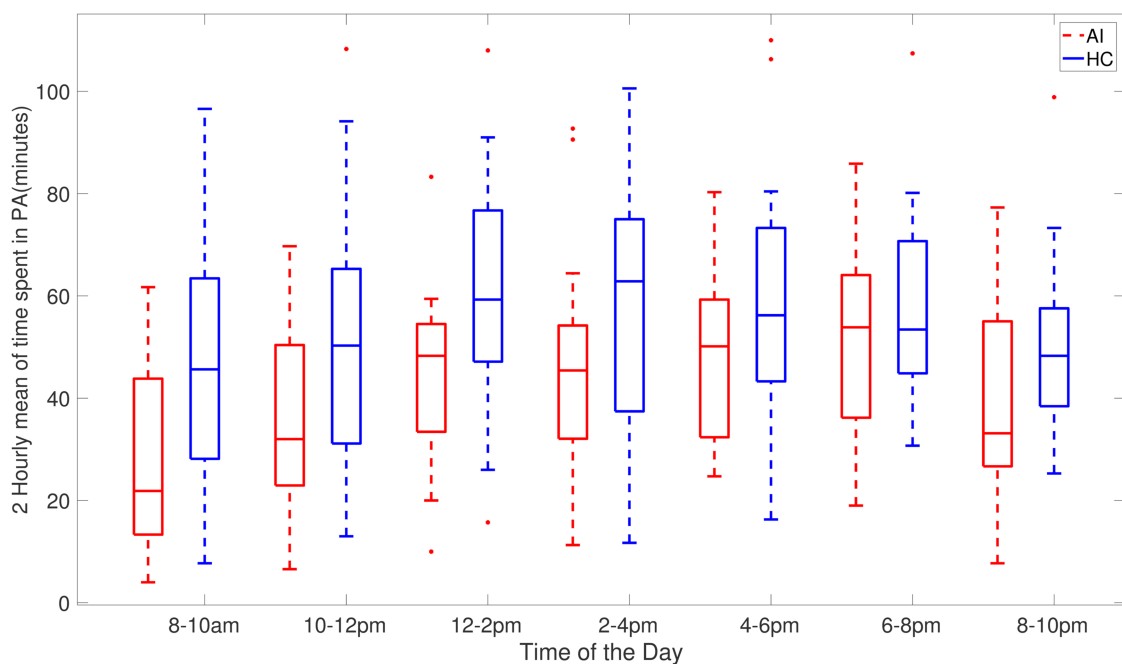

**Fig 7. Time spent in physical activities 2 hourly between 8 am to 10 pm averaged across seven days by the number of participants of each group where activity counts are greater and equal to 100.**

with insomnia. As a result, the collected activity data were not as comprehensive (e.g. did not contain the type of physical activity) as if the study were solely focused on measuring sedentary behaviour. This limitation can make it difficult to draw robust conclusions about the relationship between acute insomnia and sedentary behaviour based on the collected data. The second problem is the small size of the sample population. As most participants in the AI group are between 20 and 40 years old, we chose participants between these ages to secure a balanced (in terms of age) population and exclude the effects of ageing.

This work warrants further data collection with the specific purpose to study the sedentary behaviour. This will involve a larger population, data collected for more weekdays and weekends, as well as recordings of the specific daily activities, in addition to the sleep data, of adults with acute insomnia. The analyses of such data may further enhance the outcomes of this study.

## Conclusion

The present study is the first to analyze the patterns of sedentary activities in adults with acute insomnia using actigraphy data and compare it with same-age healthy individuals. Results from the analysis indicate that healthy control subjects are more active throughout the day in comparison with adults with acute insomnia, which is in agreement with previous studies [29, 55, 57]. Based on the results of various experiments, it can be concluded that individuals with insomnia are less active in the day than healthy individuals and the intensity of physical activities is also low in comparison to healthy adults (see Fig 7). Further, this study establishes that longer and medium-duration bouts of sedentary behaviour accumulate more statistically significant amounts of sedentary time in adults with insomnia.

## Author Contributions

**Conceptualization:** Sunita Rani, Sergiy Shelyag, Maia Angelova.

**Data curation:** Sunita Rani.

**Formal analysis:** Sunita Rani, Maia Angelova.

**Investigation:** Sunita Rani.

**Methodology:** Sunita Rani, Sergiy Shelyag, Maia Angelova.

**Software:** Sunita Rani, Sergiy Shelyag.

**Supervision:** Sergiy Shelyag, Maia Angelova.

**Validation:** Sunita Rani.

**Visualization:** Sunita Rani.

**Writing – original draft:** Sunita Rani, Maia Angelova.

**Writing – review & editing:** Sunita Rani, Sergiy Shelyag, Maia Angelova.

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
