## [Decision Letter · Decision Letter 0]

9 Jun 2023

PONE-D-23-11128Patterns of sedentary behaviour in adults with acute insomniaPLOS ONE

Dear Dr. Rani,

Thank you for submitting your manuscript to PLOS ONE. After careful consideration, we feel that it has merit but does not fully meet PLOS ONE’s publication criteria as it currently stands. Therefore, we invite you to submit a revised version of the manuscript that addresses the points raised during the review process.

We look forward to receiving your revised manuscript.

Kind regards,

Kalyana Chakravarthy Bairapareddy, PhD

Academic Editor

PLOS ONE

Journal Requirements:

5. Please ensure that you refer to Figures 1 to 9 in your text as, if accepted, production will need this reference to link the reader to the figure.

Additional Editor Comments:

Please find the comments below:

1. Recommended to rewrite the abstract with sub-headings (structured)

2. Provide clearly the selection criteria for both the groups.

3. The justification for the sample size must be provided.

4. statistical methods are repeated. Please arrange the sections properly to avoid the repetitions.

5. The study findings are not supported with proper discussion. The main objective of the study is to determine the patterns of sedentary behavior. However, these findings were not discussed properly. It is recommended to rewrite the discussion analyzing the findings (week days, time of the day, ....) and causes for sedentary behavior patterns in participants with acute Insomnia.

6. Discuss the reasons for sedentary behavior among the younger adults mentioning the other factors influencing the physical activity.

Reviewers' comments:

Reviewer's Responses to Questions

**Comments to the Author**

1. Is the manuscript technically sound, and do the data support the conclusions?

Reviewer #1: Yes

Reviewer #2: Yes

2. Has the statistical analysis been performed appropriately and rigorously? 

Reviewer #1: Yes

Reviewer #2: Yes

3. Have the authors made all data underlying the findings in their manuscript fully available?

Reviewer #1: Yes

Reviewer #2: Yes

4. Is the manuscript presented in an intelligible fashion and written in standard English?

Reviewer #1: Yes

Reviewer #2: No

5. Review Comments to the Author

Reviewer #1: Great work by authors. Keep it up. Please find my few suggestions in the attached PDF as comments.

here is an example

Page 7: Great efforts by the authors. Few suggestions from my side to make your article more understandable to the readers.

If you add the study design to the title it will help us to understand lot more information clearly in your study.

Reviewer #2: Abstract:

Kindly elaborate more in detail on quatitative results of the present study findings

Introduction:

Overall, Introduction was well structured and covered the key topic area. However, you can add a note on the various methods of screening the sedentary behavior.

Materials and methods:

The detailed structured methodology is lacking and its not well constructed and presented. Kindly rephrase the section with in detailed procedure.

6. PLOS authors have the option to publish the peer review history of their article (what does this mean?). If published, this will include your full peer review and any attached files.

Reviewer #1: No

Reviewer #2: No

---

## [Author Response · Author response to Decision Letter 0]

6 Jul 2023

Dear Sir/Madam,

We would like to thank the editor and the reviewers for their valuable time, comments, questions and suggestions. All of them were taken into account, and the manuscript has been modified accordingly. We hope we have satisfied all queries and we believe the quality of the manuscript has significantly improved and is now of sufficient quality to warrant acceptance for publication. We will also put the analysis codes and data used in the paper to an open github online repository on final acceptance of the manuscript for publication.

Our detailed responses are given below in blue. The new changes made to the manuscript are highlighted in red.

Reviewer #1: Great work by authors. Keep it up. Please find my few suggestions in the attached PDF as comments.

here is an example

Page 7: Great efforts by the authors. Few suggestions from my side to make your article more understandable to the readers.

Response/Action: 

We shifted all the table captions (table1- page 7, table2- page8, table3-page9, table4-page13, table5-page15) to the top of the tables.

If you add the study design to the title it will help us to understand lot more information clearly in your study.

Response/Action: 

1.Following the suggestion, the title has been changed.

“Patterns of sedentary behaviour in adults with acute insomnia derived from actigraphy data”

2. We have rephrased the objectives of the study. 

3. Moved text and table 1 (page 7) to the result section on page 7.

4. Table 2 - Abbreviations were explained in the caption on page 8 and captions are moved on the top of the table.

5. Table 3 - Abbreviations were explained in the caption on page 9 and captions are moved on the top of the table.

6. Table 3- sedentary bouts, total sedentary time and physical activity time is an event which are not normally distributed. To analyse these sedentary values we used the exponential distribution (discussed in materials and method section on page 5). Mean± lower/upper bound 95% confidence values, derived from the exponential distribution, are provided in Table 3 and table 4. Caption has been corrected to avoid confusion of these numbers. 

7. Figures numbers have been mentioned and corrected in the manuscript(Figure 1- page 4 ,Figure 2- page 9, Figure 3- page 10, Figure 4- page 10, Figure 5- page 12, Figure 6- page 13, Figure 7- page 14)

8. Future suggestions were added on the page 16 at the end of the Discussion section.

Reviewer #2: Abstract:

Kindly elaborate more in detail on quantitative results of the present study findings

Response/Action:

Results in the Abstract are explained in more detail on page 1.

Introduction:

Overall, Introduction was well structured and covered the key topic area. However, you can add a note on the various methods of screening the sedentary behavior.

Response/Action:

As per suggestion we have added a paragraph about the screening methods of sedentary behaviour.

“There are several methods for measuring sedentary behaviour such as self-report questionnaires, direct observation, diaries or logs, and activity trackers. During specific periods, self-report questionnaires are used to estimate the amount of time participants spend sitting or engaged in sedentary activities, whereas direct observation involves trained observers watching individuals closely and recording their sedentary behaviour. Another method is to ask participants to keep a diary or log where they can record the start and end times of their sedentary periods and any relevant information. The use of activity trackers such as pedometers or wearable devices can provide objective data regarding sedentary behaviour. In addition to measuring movement, these devices can also estimate the amount of time spent sedentary based on periods in which a person is not active. It is important to note that each screening method has its strengths and limitations, and the selection should depend on the specific research or practical purpose. In order to provide a more comprehensive assessment of sedentary behaviour, it is beneficial to integrate multiple methods, or to incorporate objective measures along with self-report measures.”

Materials and methods:

The detailed structured methodology is lacking and its not well constructed and presented. Kindly rephrase the section with in detailed procedure.

Response/Action:

Several changes to Materials and Methods section and Results section were made to address the comment. The updated version includes a structured approach with subtitles for the different methods. Further details were added to selection and justification of the sample size. Table 1 and relevant text were moved to Results section.

EDITOR:

Please find the comments below:

1. Recommended to rewrite the abstract with sub-headings (structured)

Response/Action:

The Abstract has been rewritten in a structured way with sub-headings.

2. Provide clearly the selection criteria for both the groups.

3. The justification for the sample size must be provided.

Response/Action:

Selection criteria and justification of the sample size are provided on page 4.

4. Statistical methods are repeated. Please arrange the sections properly to avoid the repetitions.

Response/Action:

The statistical methods are rearranged (Sections Materials and Methods and Results) in order to avoid repetition.

5. The study findings are not supported with proper discussion. The main objective of the study is to determine the patterns of sedentary behavior. However, these findings were not discussed properly. It is recommended to rewrite the discussion analyzing the findings (week days, time of the day, ....) and causes for sedentary behavior patterns in participants with acute Insomnia.

Response/Action:

Study findings are explained in the Discussion section in paragraph 4 and 5 on page 15 and 16.

6. Discuss the reasons for sedentary behavior among the younger adults mentioning the other factors influencing the physical activity.

Response/Action:

Some possible reasons for sedentary behaviour are given in paragraph 1 of Discussion section on page 15. However, as the dataset had only actigraphy recordings and did not contain any information about the daily physical activities, it was not possible to make robust conclusions about the reasons for the sedentary behaviour based entirely on the actigraphy recordings.

---

## [Editor Report · Decision Letter 1]

2 Aug 2023

PONE-D-23-11128R1Patterns of sedentary behaviour in adults with acute insomnia derived from actigraphy dataPLOS ONE

Dear Dr. Rani, 

Thank you for submitting your manuscript to PLOS ONE. After careful consideration, we feel that it has merit but does not fully meet PLOS ONE’s publication criteria as it currently stands. Therefore, we invite you to submit a revised version of the manuscript that addresses the points raised during the review process.

Please submit your revised manuscript by Sep 16 2023 11:59PM. If you will need more time than this to complete your revisions, please reply to this message or contact the journal office at plosone@plos.org. Please include the following items when submitting your revised manuscript:A rebuttal letter that responds to each point raised by the academic editor and reviewer(s). You should upload this letter as a separate file labeled 'Response to Reviewers'.A marked-up copy of your manuscript that highlights changes made to the original version. You should upload this as a separate file labeled 'Revised Manuscript with Track Changes'.An unmarked version of your revised paper without tracked changes. You should upload this as a separate file labeled 'Manuscript'.If applicable, we recommend that you deposit your laboratory protocols in protocols.io to enhance the reproducibility of your results. Protocols.io assigns your protocol its own identifier (DOI) so that it can be cited independently in the future. For instructions see: https://journals.plos.org/plosone/s/submission-guidelines#loc-laboratory-protocols. Additionally, PLOS ONE offers an option for publishing peer-reviewed Lab Protocol articles, which describe protocols hosted on protocols.io. Read more information on sharing protocols at https://plos.org/protocols?utm_medium=editorial-email&utm_source=authorletters&utm_campaign=protocols.

We look forward to receiving your revised manuscript.

Kind regards,

Kalyana Chakravarthy Bairapareddy, PhD

Academic Editor

PLOS ONE

Journal Requirements:

**Additional Editor Comments:**

Language correction is required for the manuscript. Please double-check the grammar. Please ensure that each sentence in the abstract begins with a capital letter.

---

## [Author Response · Author response to Decision Letter 1]

15 Aug 2023

Response/Action: 

We have reviewed all the references for the retracted articles. There are no retracted articles cited in the manuscript and the references are formatted according to the journal's requirements.

Additional Editor Comments:

Language correction is required for the manuscript. Please double-check the grammar. Please ensure that each sentence in the abstract begins with a capital letter.

Response/Action: 

We have reviewed the manuscript for grammar errors and ensured that every sentence in the abstract starts with a capital letter. The figures comply with the journal's requirements.

1. In the Methods section please include the informed consent statement to reflect whether "written or verbal" informed consent was obtained from all participants for inclusion in the study.

Response/Action: 

We have reviewed and added the information about consent in the Method section :

” Informed written consent was obtained from all participants during the original data collection.”

---

## [Editor Report · Decision Letter 2]

22 Aug 2023

Patterns of sedentary behaviour in adults with acute insomnia derived from actigraphy data

PONE-D-23-11128R2

Dear Dr. Sunitha,

We’re pleased to inform you that your manuscript has been judged scientifically suitable for publication and will be formally accepted for publication once it meets all outstanding technical requirements.

Kind regards,

Kalyana Chakravarthy Bairapareddy, PhD

Academic Editor

PLOS ONE

---

## [Editor Report · Acceptance letter]

8 Sep 2023

PONE-D-23-11128R2 

Patterns of sedentary behaviour in adults with acute insomnia derived from actigraphy data 

Dear Dr. Rani:

I'm pleased to inform you that your manuscript has been deemed suitable for publication in PLOS ONE. Congratulations! Your manuscript is now with our production department. 

Kind regards, 

on behalf of

Dr. Kalyana Chakravarthy Bairapareddy 

Academic Editor

PLOS ONE